# Offline Contextual Bayesian Optimization

**Ian Char**[1], **Youngseog Chung**[1], **Willie Neiswanger**[1], **Kirthevasan Kandasamy**[2], **Andrew Oakleigh Nelson**[3], **Mark D Boyer**[3], **Egemen Kolemen**[3], and **Jeff Schneider**[1]

[1]Department of Machine Learning, Carnegie Mellon University
*{ichar, youngsec, willie, schneide}@cs.cmu.edu*
[2]Department of EECS, University of California Berkeley
*kandasamy@eecs.berkeley.edu*
[3]Princeton Plasma Physics Laboratory
*{anelson, mboyer, ekolemen}@pppl.gov*

## Abstract

In black-box optimization, an agent repeatedly chooses a configuration to test, so as to find an optimal configuration. In many practical problems of interest, one would like to optimize several systems, or "tasks", simultaneously; however, in most of these scenarios the current task is determined by nature. In this work, we explore the "offline" case in which one is able to bypass nature and choose the next task to evaluate (e.g. via a simulator). Because some tasks may be easier to optimize and others may be more critical, it is crucial to leverage algorithms that not only consider which configurations to try next, but also which tasks to make evaluations for. In this work, we describe a theoretically grounded Bayesian optimization method to tackle this problem. We also demonstrate that if the model of the reward structure does a poor job of capturing variation in difficulty between tasks, then algorithms that actively pick tasks for evaluation may end up doing more harm than good. Following this, we show how our approach can be used for real world applications in science and engineering, including optimizing tokamak controls for nuclear fusion.

## 1 Introduction

Black-box optimization is the problem in which one tries to find the maximum of an unknown function solely using evaluations for specified inputs. In many interesting scenarios, there is a collection of unknown, possibly correlated functions (or *tasks*) that need to be simultaneously optimized. This problem set up often occurs in applications where one wants to design an agent that makes an action based on some contextual information from the environment. However, we would prefer that the agent not run potentially costly or poor performing experimental actions online. Also, because the agent may have to make these decisions at a rapid pace, we often do not have time to compute an expensive experimentation policy. We consider applications that provide the ability to run offline experiments where nature can be bypassed and the contextual information can be manually set (e.g. on a surrogate system or on a simulation). These experiments are used to discover a good action policy which is then encoded into a fast cache, such as a look-up table. Even though the experiments are done offline, they are still expensive and we must search the design space efficiently. The following are examples of this problem:

- **Nuclear Fusion** A tokamak is a device used to magnetically confine plasma and is the most commonly pursued means of generating power from controlled nuclear fusion. A current obstacle in realizing sustained nuclear fusion is the difficulty in maintaining the plasma's stability at the required temperatures and pressures for a prolonged period of time. We

consider the stability of the plasma as an output to optimize, where the input is the controls for the tokamak. The optimal action depends on the current state of the plasma, so each plasma state can be regarded as its own task to optimize. We cannot search for a good control policy during live experiments because of cost, limited time available on the device, and the need to provide a real-time controller that operates in a millisecond scale control loop. However, we do have a simulation (forward model) that may be used with Bayesian optimization offline to discover a good controller. Importantly, the simulator allows one to manually set the current state of the plasma, and thus prudently selecting states to optimize over becomes an important part of the problem.

- **Database Tuning** Consider the problem of tuning the configuration of a database so as to minimize the latency, the CPU/memory footprint, or any other desired criteria. The performance of a configuration depends critically on the underlying hardware and the workload [Van Aken et al., 2017]. Since these variables can change when databases are deployed in production, we need to simultaneously optimize for these different tasks.

In each of the above settings, difficulty of the tasks may vary drastically. For example, in the nuclear fusion application, if the current state of the plasma is already stable, the stability may be less sensitive to controls, leading to an easy optimization landscape. On the other hand, when the plasma is in an unstable state, it may be that only a small set of controls will lead to improved stabilization and finding them may require many more experiments.

In this paper, we propose a Thompson sampling approach for adaptively picking the next task and input for evaluation. Unlike other Bayesian Optimization (BO) algorithms, evaluations are picked in order to efficiently estimate the optimal action for *each* task, where these optimal actions are most likely distinct. This algorithm comes with theoretic guarantees, and we show that it often enjoys a significant boost in performance when compared to uniformly distributing resources across tasks and other state of the art methods. Another contribution of this paper is showing the significance of model choice in this setting. We argue that when using a single Gaussian process (GP) to jointly model correlated tasks, the choice of kernel is crucial for estimating the difficulty of each task. We believe that model selection here is even more important than in single-task BO because incorrect estimates can lead to poorly managed resource allocation for tasks. We give an example where inaccurately modeling reward structure between tasks via a stationary kernel severely hurts our algorithm. Following this, we suggest a kernel with a lengthscale that varies with tasks and show that this more intelligent kernel again allows our algorithm to enjoy a performance boost. An implementation of our algorithm and synthetic experiments can be found at `https://github.com/fusion-ml/OCBO`.

We end this paper by showing an application of our method to the nuclear fusion problem. In particular, we optimize tokamak controls for a set of different plasma states using a tokamak simulator. We observe that our method is able to identify where best to devote resources, leading to efficient optimization.

## 2 Related Work

Our algorithm falls under the general umbrella of Bayesian optimization [Shahriari et al., 2015, Frazier, 2018]. As is common in BO, we use a GP prior to guide us in selecting next evaluations to make. Previously, in the context of active learning and active sensing, techniques have been made that use GPs to select the most informative points for evaluation [Pasolli and Melgani, 2011, Seo et al., 2000, Guestrin et al., 2005]. In contrast, our goal is optimization which is more in line with bandit methods. Under the bandits setting, Srinivas et al. [2009] use an upper confidence bound approach with GPs and show that such a strategy results in sublinear cumulative regret. As an alternative to the upper confidence bound approach, Russo and Van Roy [2014] show that one can achieve sublinear cumulative regret using a posterior sampling (or Thompson sampling) approach. The method we present here is also a posterior sampling method, and it falls into the general framework of myopic posterior sampling described by Kandasamy et al. [2019a].

Our setting is related to online *contextual* bandits [Krause and Ong, 2011, Agrawal and Goyal, 2013, Auer, 2002], where each task can be viewed as a different context. In these earlier works, the agent chooses an action online for a context that is chosen by the environment. In our setting, we wish to find the optimal action offline in advance and can choose the contexts we invest our experimentation effort on. The models in the works of Krause and Ong [2011], Swersky et al. [2013] are of particular

interest. Both works use a GP to jointly model correlated contexts and propose a similar structure for the joint GP's kernel. We adopt a similar strategy, however, our model has the advantage that lengthscales can vary between contexts.

A similar contextual optimization problem shows up in reinforcement learning (RL). While the common RL setup has contexts delivered solely by the environment, there is some work on actively choosing contexts [Fabisch and Metzen, 2014, Fabisch et al., 2015]. This work proposes methods for approximating the expected improvement (EI) in the overall objective. Similarly, the objective can be written in terms of entropy and experiments may be chosen in terms of its expected improvement [Metzen, 2015, Swersky et al., 2013]. In our empirical study, we compare to expected improvement for task and action selection.

Unlike many other problems under the BO setting, our algorithm searches for an optimal action for each task rather than a single optimal action. This serves as a contrasting feature from other problems in multi-task BO [Swersky et al., 2013, Toscano-Palmerin and Frazier, 2018], in which a single action that performs optimally across all objectives simultaneously is sought. The works most similar to ours present algorithms based around EI or knowledge gradients [Frazier et al., 2009]. In particular Ginsbourger et al. [2014] and Pearce and Branke [2018] consider the same problem setting, but focus on the case where the set of tasks is continuous. Although our algorithm can be adapted to this case, we focus on the finite task setting and show that our posterior sampling approach provides a theoretically-grounded, competitive alternative. We also note that previous works have used RBF kernels for their synthetic experiments, and while this adequately models the reward landscape for their relatively smooth functions, we claim that when there is a large variation in task difficulty this may cause these algorithms to do more harm than good.

## 3 Thompson Sampling for Multi-Task Optimization

### 3.1 Preliminaries

For the following, let $\mathcal{X}$ be the collection of tasks and let $\mathcal{A}$ be the compact set of possible actions. Throughout this work, we assume that the same set of actions is available for each task. Let $f : \mathcal{X} \times \mathcal{A} \to \mathbb{R}$ be the bounded reward function, where $f(x, a)$ is the reward for performing action $a$ in task $x$. Let $\hat{h} : \mathcal{X} \to \mathcal{A}$ be our estimated mapping from task to action. Our goal is then to find such an $\hat{h}$ which maximizes the following objective:

$$\sum_{x \in \mathcal{X}} f\left(x, \hat{h}(x)\right) \omega(x) \tag{1}$$

where $\omega(x) \geq 0$ is some weighting on $x$ that may depend on the probability of seeing $x$ at evaluation time or the importance of $x$. We usually assume that $\mathcal{X}$ is finite; however, we also consider the case when $\mathcal{X}$ is continuous in Appendix D, in which case the sum in (1) becomes an integral. At round $t$ of optimization, we pick a task $x_t$ and an action $a_t$ to perform a query $(x_t, a_t)$ and observe a noisy estimate of the function $y_t = f(x_t, a_t) + \epsilon_t$, where $\epsilon_t \sim N(0, \sigma_\epsilon^2)$ and is iid. Let $D_t$ be the sequence of queried tasks, actions, and rewards up to time $t$, i.e. $D_t = \{(x_1, a_1, y_1,), \ldots, (x_t, a_t, y_t)\}$. Additionally, define $\hat{y}_t(x)$ to be the best reward observed for task $x$ up to time $t$, $\hat{a}_t(x)$ to be the action made to see this corresponding reward, and $\mathcal{A}_t(x)$ to be the set of all actions made for task $x$ up to time $t$.

In this work, we assume that $f$ is drawn from a *Gaussian process* (GP) prior. A GP is characterized by its mean function, $\mu(\cdot)$ and kernel (or covariance) function $\sigma(\cdot, \cdot)$. Then for any finite set of variables, $z_1, \ldots, z_n \in \mathcal{X} \times \mathcal{A}$, $[f(z_1), \ldots, f(z_n)]^T \sim N(m, \Sigma)$, where $m \in \mathbb{R}^n$, $\Sigma \in \mathbb{R}^{n \times n}$, $m_i = \mu(z_i)$, and $\Sigma_{i,j} = \sigma(z_i, z_j)$. It is important to note that by selecting different kernel functions we make implicit assumptions about the smoothness of $f$. A valuable property of the GP is that its posterior is simple to compute. We denote $\mu_t$ and $\sigma_t$ to be the posterior mean and posterior kernel functions after seeing $t$ evaluations. For more information about GPs see Rasmussen and Williams [2005].

### 3.2 Multi-Task Thompson Sampling

We now describe our proposed algorithm called Multi-Task Thompson Sampling (MTS), which is presented in Algorithm 1 for the case in which $\mathcal{X}$ is a finite set of correlated tasks. The algorithm is

an extension of Thompson sampling [Thompson, 1933] to the multi-task setting. Simply put, MTS acts optimally with respect to samples drawn from the posterior. That is, at every round a sample for the reward function is drawn, and this sample is used as if it was ground truth to identify the task in which the most improvement can be made. After doing this for $T$ iterations, we return the estimated mapping $\hat{h}$ such that $\hat{h}(x) = \hat{a}_T(x)$ if an evaluation was made for task $x$; otherwise, $\hat{h}(x)$ maps to an $a \in \mathcal{A}$ drawn uniformly at random. Note that when tasks are assumed to be independent, Algorithm 1 can be modified by instead using a separate GP prior for each task and drawing samples from each at every iteration.

---

**Algorithm 1** Multi-Task Thompson Sampling (MTS)

---

**Input:** capital $T$, initial capital $t_{init}$, mean function $\mu$, kernel function $\sigma$.
Do random search on tasks in round-robin fashion until $t_{init}$ evaluations are expended.
**for** $t = t_{init} + 1$ **to** $T$ **do**
    Draw $\widetilde{f} \sim GP(\mu, \sigma)|D_{t-1}$.
    Set $x_t = \underset{x \in \mathcal{X}}{\operatorname{argmax}} \left[ \left( \max_{a \in \mathcal{A}} \widetilde{f}(x, a) - \max_{a \in \mathcal{A}_t(x)} \widetilde{f}(x, a) \right) \omega(x) \right]$.
    Set $a_t = \underset{a \in \mathcal{A}}{\operatorname{argmax}} \widetilde{f}(x_t, a)$.
    Observe $y_t = f(x_t, a_t)$.
    Update $D_t = D_{t-1} \cup \{(x_t, a_t, y_t)\}$.
**end for**
**Output:** $\hat{h}$

---

One benefit of this algorithm is that it comes with theoretic guarantees. For the following, define $a_t^*(x)$ to be the past action played for task $x$ that yields the largest expected reward. That is,

$$
a_t^*(x) := \begin{cases} \underset{a \in \mathcal{A}_t(x)}{\operatorname{argmax}} f(x, a) & \mathcal{A}_t(x) \neq \emptyset \\ \underset{a \in \mathcal{A}}{\operatorname{argmin}} f(x, a) & \text{else} \end{cases}
$$

Note that $a_t^*(x)$ has an implicit dependence on $f$.

**Theorem 1.** *Define the maximum information gain to be $\gamma_T := \max_{D_T} I(D_T; f)$, where $I(\cdot; \cdot)$ is the Shannon mutual information. Assume that $\mathcal{X}$ and $\mathcal{A}$ are finite. Then if Algorithm 1 is played for $T$ rounds where $t_{init} = 0$*

$$
\mathbb{E}\left[R_{T,f}\right] \leq |\mathcal{X}| \left( \frac{1}{T} + \sqrt{\frac{|\mathcal{X}||\mathcal{A}|\gamma_T}{2T}} \right)
$$

*where the expectation is with respect to the data sequence collected and $f$, and where $R_{T,f}$ is defined to be*

$$
R_{T,f} := \frac{\sum_{x \in \mathcal{X}} \omega(x) \left( \max_{a \in \mathcal{A}} f(x, a) - f(x, a_T^*(x)) \right)}{\sum_{x \in \mathcal{X}} \omega(x) \left( \max_{a \in \mathcal{A}} f(x, a) - \min_{a \in \mathcal{A}} f(x, a) \right)}
$$

*when the denominator is not $0$. Otherwise, $R_{T,f}$ takes the value of $0$.*

The proof of this theorem (see Appendix A) uses ideas from Kandasamy et al. [2019a]. This result gives a bound on the expected normalized *total simple regret*. Here, *simple regret* is the difference between the best reward and the best reward for a played action (i.e. $\max_{a \in \mathcal{A}} f(x, a) - f(x, a_T^*(x))$), and *total simple regret* refers to the simple regret summed across all tasks. The $\sqrt{|\mathcal{X}||\mathcal{A}|}$ factor in the theorem accounts for the number of actions that can be taken at every step, and the $\sqrt{\gamma_T}$ factor characterizes the complexity of the prior over the tasks. We suspect that our proof technique may have lead to a somewhat loose bound because there is an extra dependence of $|\mathcal{X}|$; that being said, we still get that the rate of decrease is dominated by $\sqrt{\frac{\gamma_T}{T}}$, which is the same as the single-task regret rate [Russo and Van Roy, 2014].

An important implication of this result is that there is no task in which we will have especially bad results, and when $\gamma_T = o(T)$, the normalized simple regret converges to 0 in expectation for every task. We note that Srinivas et al. [2009] give bounds on the maximum information gain for a single

GP in a few standard cases. For example, when dealing with a GP over a $d$-dimensional compact set using an RBF kernel, $\gamma_T^{(RBF)} = \mathcal{O}(\log(t)^{d+1})$. Finally, we note that these types of results can usually be generalized to infinite action spaces via known techniques [Russo and Van Roy, 2016, Bubeck et al., 2011]

**Continuous task setting.** Often one is confronted by a set of tasks that are correlated and continuous. The problem of finding a policy in this setting is inherently different because the tasks seen offline will not be the exact same as the tasks encountered when the policy is deployed. Nevertheless, MTS can be adapted to this setting by leveraging the posterior mean instead of $\mathcal{A}_t(x)$ (details are in Appendix D). Even though greedily picking evaluations to increase improvement within a single task is likely not optimal here, we found that our algorithm performs competitively with other more expensive state of the art methods, especially in higher dimensional settings.

### 3.3 Synthetic Experiments

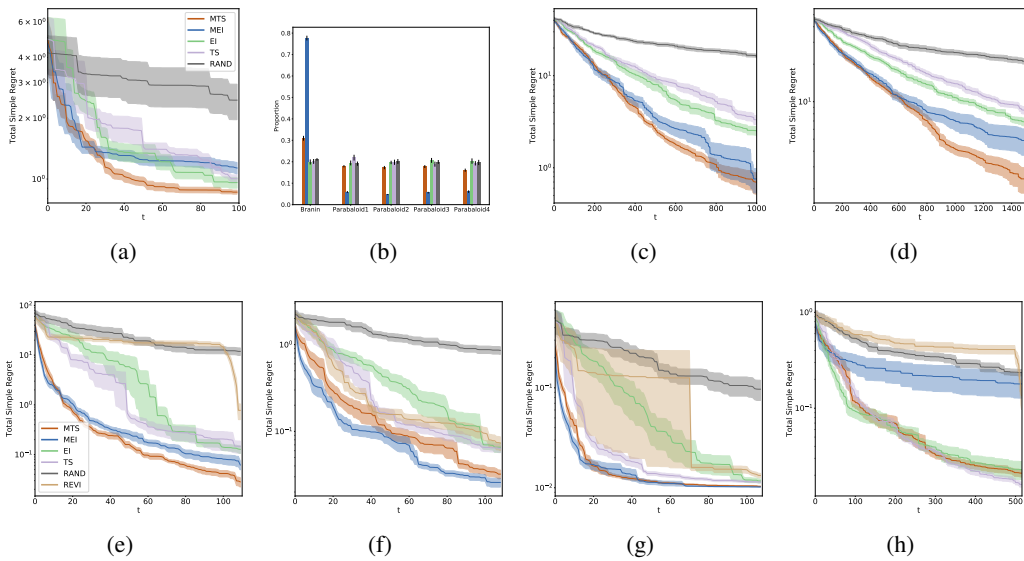

Figure 1: **Synthetic experiments for MTS**. Each of the experiments were averaged over 10 trials and show the mean value and standard error. The plots for independent tasks in the top row are as follows: **(a)** Total simple regret when tasks are Branin-Hoo and four other identical 2D parabaloids, **(b)** the corresponding proportion of capital spent for each task in (a), **(c)** total simple regret for 30 random 4D functions, **(d)** total simple regret for 30 random 6D functions. The second row shows total simple regret for Branin 1-1 **(e)**, Hartmann 2-2 **(f)**, Hartmann 3-1 **(g)**, and Hartmann 4-2 **(h)**. In many of the cases, we must estimate the true optimal value and therefore cannot plot the true regret (see Appendix F).

**Independent Tasks**. For this setting we compare MTS against a suite of baselines which distribute resources evenly amongst tasks. The first of which, selects task-action pairs uniformly at random. Additionally, we compare against the procedure of selecting a task uniformly at random and applying standard Thompson sampling (TS) or expected improvement (EI) at every iteration. This is essentially equivalent to iteratively optimizing for each task using standard BO methods, using the same amount of computation for each task. Additionally, we compare against the procedure described by Swersky et al. [2013] in Section 3.2. This algorithm, which we call Multi-task Expected Improvement (MEI), picks the task with the greatest expected improvement at every iteration. However, we do not impute missing data across tasks using the posterior mean (in the independent case it is impossible to do so). Although the setting this algorithm was designed for is slightly different (it is assumed there is one optimal action over all tasks), the approach is still applicable to our setting. The following experiments are averaged over 10 trials. We start by evaluating each task with 5 points drawn uniformly at random. Each task is modeled by a GP with an RBF kernel, and hyperparameters are tuned for a GP every time an observation is seen for its corresponding task. For two-dimensional

functions, hyperparameters are tuned according to marginal likelihood, but for greater dimensions, tuning is done using a blend of marginal likelihood and posterior sampling. This method was found to be more robust by Kandasamy et al. [2019b]. Here, and throughout this section, we leverage the Dragonfly library for our experiments [Kandasamy et al., 2019b]. Lastly, in every experiment we let $\omega(x) = 1$ for all $x \in \mathcal{X}$ and give noiseless feedback to the algorithms.

For the first synthetic problem, we wish to optimize over 5 functions: four of which are concave parabaloids (with a range of $[0, 1]$) and the other being the Branin-Hoo function [Branin, 1972]. Not only does the Branin-Hoo function have a greater scale, but it is also much more complex. Thus, one might imagine that virtually all resources should be invested in optimizing this function, which is the behavior displayed by MEI. However, we see that MTS performs best by distributing resources more liberally amongst tasks (see Figure 1 (a) and (b)). We also test these methods on 30 randomly generated functions in four and six dimensions (see Appendix B for details), and we found MTS to be the strongest performer.

**Correlated Finite Tasks**. To evaluate our method in the correlated finite task setting, we take multi-dimensional functions, treat the first few dimensions as task space, and select equispaced tasks in this space to focus on. In particular we use the Branin-Hoo, Hartmann 4, and Hartmann 6 [Picheny et al., 2013] function to create Branin 1-1, Hartmann 2-2, Hartmann 3-1, and Hartmann 4-2, where the first number is the task dimension and the second is the action dimension. We consider 10, 9, 8, and 16 tasks for each of these functions, respectively. The set up is identical to before except for that a single GP is used to jointly model tasks, and the GP is tuned by maximizing the marginal likelihood for all experiments. In addition to the previous baselines, we also compare against the REVI algorithm introduced by [Pearce and Branke, 2018]. This algorithm, based on knowledge gradients Frazier et al. [2009], picks task-action pairs for evaluation by estimating which will increase the GP mean the greatest across all tasks. That is, it myopically tries to optimize (1) at each round by using the GP mean as a proxy. In the risk-averse setting in which the policy returned maps task to the best action seen throughout training (i.e. the setting we have considered throughout this paper), the authors recommend running EI in a round-robbin fashion at the end of training. As such, we end the optimization with one round of EI for REVI.

The results are shown in the second row of Figure 1. We also compare in the risk-neutral setting (i.e. when the policy is derived from posterior mean) in Appendix C. For the majority of the cases MTS and MEI are the best performers. The exception to this is the experiment done on the Hartmann 4-2 function. Here, MEI does significantly worse than standard EI and TS methods, while MTS has about the same performance. We found that MEI focuses almost all of its capital on just three tasks, which most likely causes the poor performance. In all cases, MTS and MEI outperform REVI, even when a round of EI is performed at the end of execution. We believe that REVI does not perform as well in these experiments since the tasks considered are spread out in task space, and REVI focuses less on tasks at the boundary of the space (see Appendix E for visualizations). Indeed, if we consider continuous correlated tasks instead (see Appendix D), REVI becomes a strong performer. With that being said, we argue that the formulation of these experiments is natural for real life applications, and the set up for our fusion experiments in Section 5 is similar to this.

# 4   Modeling Variation in Difficulty

The selection of hyperparameters for the kernel function of a GP is often key to whether the landscape can be modeled well. Usually these hyperparameters include *lengthscale*, which determines how correlated points are based on their distance to each other, and *scale*, which determines the magnitude of correlation. Intuitively, these values provide some indication of the optimization landscape's difficulty. For example, larger lengthscales imply more smooth functions, which are often easier to optimize for. From a more theoretical standpoint, the hyperparameters have a direct effect on the maximum information gain and therefore impact regret bounds shown by Theorem 1 and Russo and Van Roy [2014].

Intuitively, hyperparameters should vary between tasks in order to adequately model any difference in difficulty between them. One method for achieving this when jointly modelling tasks is via a locally stationary kernel, i.e. hyperparameters vary with respect to tasks but not with actions. Although there may be many ways to achieve this, a straightforward approach is to use the Gibbs kernel [Gibbs, 1998]. The Gibbs kernel is a non-stationary variant of the RBF kernel that allows the lengthscale and

scale to vary over the space. Where $z, z' \in \mathcal{X} \times \mathcal{A}$, $P_X = \dim(\mathcal{X})$ and $P_A = \dim(\mathcal{A})$,

$$\sigma(z, z') = \prod_{p=1}^{P_X + P_A} \left[ \sqrt{\frac{2\ell_p(z)\ell_p(z')}{\ell_p^2(z) + \ell_p^2(z')}} \exp\left( \frac{-(z_p - z_p')^2}{\ell_p^2(z) + \ell_p^2(z')} \right) \right] \tag{2}$$

Here, $\ell_p$ is the non-negative *lengthscale function* that characterizes the hyperparameters for the $p^{th}$ dimension. The above can be separated into the product of a kernel over the task space and a kernel over the action space, i.e. $\sigma(z, z') = \sigma_X(x, x')\sigma_A(z, z')$ where $z = (x, a)$ and $z' = (x', a')$. To suit our needs, we make all lengthscale functions for $\sigma_X$ constant functions so that $\ell_i(z) = \ell_i$ where $\ell_i \in \mathbb{R}$ and $\ell_i > 0$ for all $i = 1, \ldots, P_X$. As for $\sigma_A$, we limit the lengthscale functions to only depend on the task component of $z$. Altogether,

$$\sigma(z, z') = \sigma_X(x, x')\sigma_A(z, z') \tag{3}$$

$$= \left( \prod_{i=1}^{P_X} \exp\left( \frac{-(x_i - x_i')^2}{2\ell_i^2} \right) \right) \left( \prod_{j=1}^{P_A} \sqrt{\frac{2\ell_j(x)\ell_j(x')}{\ell_j^2(x) + \ell_j^2(x')}} \exp\left( \frac{-(a_j - a_j')^2}{\ell_j^2(x) + \ell_j^2(x')} \right) \right) \tag{4}$$

Note that with this modification $\sigma_X$ reduces to the RBF kernel. Furthermore, for any fixed task $x \in \mathcal{X}$ (i.e. we only consider $z = (x, a)$, $z' = (x, a')$) the entire kernel reduces to the RBF kernel. As such, we are left with a locally stationary kernel, where the hyperparameters only vary as the task varies. In the proceeding section, we leverage this model with our posterior sampling methods.

**Synthetic Example**. For the correlated task experiment in Section 3.3, tasks are generally quite similar, so MTS and MEI can do well when the GP uses an RBF kernel. However, we now wish to optimize 10 correlated tasks of variable difficulty. To create the tasks, we take slices from the function visualized in Figure 2 (see Section B in Appendix for details). Like many real-world tasks, this function has areas that make for an interesting optimization problem and others that are quite boring. In order to optimize well, we use the kernel presented in (3), where the lengthscale function of each action dimension is the soft plus of a quadratic polynomial, and the coefficients of each polynomial are treated as hyperparameters. We form a hierarchical probabilistic model by placing Normal priors over each hyperparameter. Then, for every iteration of our algorithm, we now make decisions according to a posterior sample drawn from this hierarchical model. For our implementation of these models, we use probabilistic programming and BO frameworks [Carpenter et al., 2017, Neiswanger et al., 2019].

In practice, this does a superior job at modeling each task. To show this, each of the ten tasks were evaluated at five points. Then, both our suggested model and a stationary model using an RBF kernel was fit to the data. The difference becomes especially clear when looking at tasks that are relatively flat functions, since the stationary GP falsely estimates large peaks. This can be especially damaging in our case where we select tasks based on possible performance improvements.

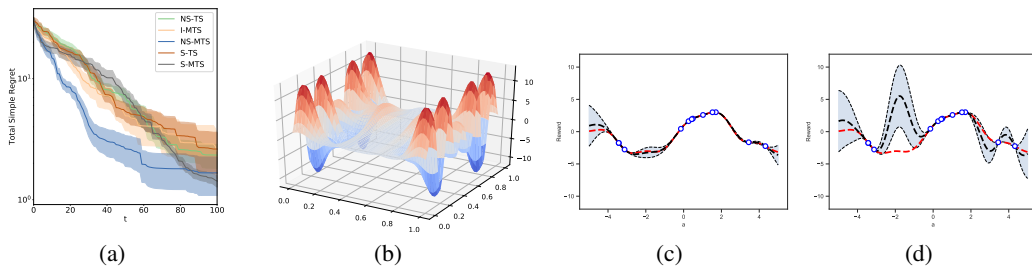

|       |       |       |       |
|-------|-------|-------|-------|
| (a)   | (b)   | (c)   | (d)   |

Figure 2: **Tasks of varying difficulty**. **(a)**: Average best seen rewards summed across all tasks of varying difficulty. Each curve is averaged over 12 trials, and the shaded region shows the standard error. **(b)**: Surface of the function used to generate correlated tasks. **(c)** shows our proposed model on an easy task, and **(d)** shows a stationary model on the same easy task. Here, the red line shows the true function, the black line shows the posterior mean, the blue points show evaluations made for the corresponding task, and the shaded area shows high confidence regions.

We run optimization using MTS and standard Thompson sampling where tasks are picked uniformly at random. Moreover, we run these algorithms using the model described above, using a single GP

that jointly models tasks using an RBF kernel, and using several GPs, each corresponding to a task and using an RBF kernel (i.e. assume that tasks have no correlation). In all cases, we use posterior sampling to select hyperparameters. For simplicity, we append prefixes to these methods where "I" stands for independent GPs, "S" stands for stationary GP, and "NS" stands for non-stationary GP. The results in Figure 2 show that one can be negatively affected by picking tasks given an ill-suited model. Although S-MTS ultimately ends up performing well, it initially struggles when compared to S-TS and I-MTS. Thus depending on resources available, it may be better to either forego shared information to better model the function or distribute resources to tasks uniformly. That being said, disregarding both shared information and picking tasks intelligently, as in I-TS, results in the worst performance (not pictured here). Notice that when tasks can be modeled appropriately, distributing resources according to our algorithm is again beneficial as shown by NS-MTS.

# 5    Application to Nuclear Fusion

Nuclear fusion is regarded as the energy of the future since it presents the possibility of unlimited clean energy. The most widespread method of realizing fusion reactions requires heating up isotopes of hydrogen to temperatures of hundreds of millions of degrees using a magnetic device called a *tokamak*. In this state, the nuclei of two nearby atoms may overcome electrostatic repulsion force between them to form a single nucleus, releasing energy. One obstacle in utilizing fusion as a feasible energy source, however, is the stability of the reaction. Once the plasma has reached a reaction state, it is uncertain how the tokamak controls should be modified to address the varying state of the plasma in order to sustain the fusion reaction. We tackle this problem by attempting to learn optimal controls offline via a simulator. In particular, we apply our algorithm to determine a mapping from plasma state to tokamak neutral beam controls.

**Experiment Set Up**. We consider a collection of 7 tasks that represent different plasma states. An evaluation of an action on a task corresponds to setting the tokamak beam controls and conducting a simulation on the selected state of the plasma. These simulations are run on the predictive mode of TRANSP, which simulates tokamaks. Both the action space and the task space are two dimensional, and the reward is a weighted sum of plasma stability and fusion reaction efficiency. Appendix G.1 provides more details of this experiment.

We compare the performance of 4 algorithms: MTS and standard Thompson sampling with a joint GP model across both states (tasks) and actions (denoted J-MTS and J-TS, respectively), and MTS and standard Thompson sampling with independent GP models across actions for each state (denoted I-MTS and I-TS). Because we had no reason to believe that there will be a drastic difference in difficulty between tasks, we used a non-stationary kernel for these initial experiments. Moreover, the experimental settings were identical to the two-dimensional synthetic experiments in Section 3.3, except for 2 differences: 5 trials of each algorithms were run with each trial consisting of 200 evaluations, and in each trial, we allow up to 10 evaluations to be run in parallel. We rely on parallel optimization here since each query has high simulation overhead ($> 1$ hour per simulation experiment). For more details regarding the setup for the fusion simulation experiments, see Appendix G.1.

Over 200 evaluations, we observe that J-MTS (the blue curve in Figure 3 (a)) outperforms J-TS, which shows the merit of focusing on states that are deemed more "difficult", rather than uniformly selecting a state. This behavior can also be seen in the performance and query plots per task in Figure 3 (b). Once the reward has levelled off in a certain task (e.g. plasma state 3, 4), J-MTS stops querying the task and queries other tasks that are predicted to provide improvement, while J-TS will still query the task as it chooses tasks randomly. This algorithm also outperforms the MTS and TS with independent models for each state, I-MTS and I-TS, which shows the merit of jointly learning the state-action space and sharing information across the correlated states. With independent GPs for each state, I-TS outperformed I-MTS. We believe this may be because of occasional erratic outputs from the simulator. This occurred more frequently in some states than in others and when the limits of the simulator were tested with extreme controls (e.g. very high power for all neutral beams). In such cases, I-MTS will estimate the reward landscape to be non-smooth and focus on the particular state. This behavior is shown by the high proportion of queries made by I-MTS in state 3 (Figure 3 (c)) despite little further improvement in reward (Figure 3 (b)). It is worthwhile to note, however, that this behavior is not evident in J-MTS, and we believe that using all queries of state-action pairs to learn a single joint state-action model is more robust to extreme observations from a particular

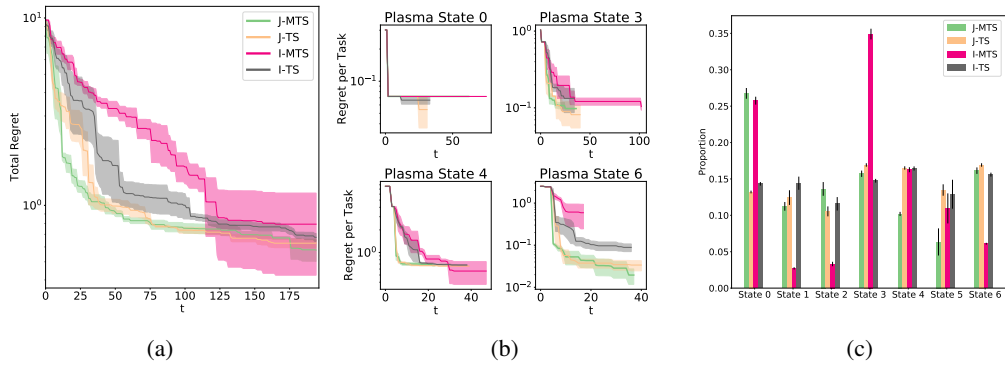

Figure 3: **Fusion Simulation Experiments.** Each of the above show average values and standard error from 5 trials. **(a)** shows the total regret summed across all tasks, **(b)** the regret achieved in each task, and **(c)** the proportion of capital spend in each task. Note that curves differ in length for (b) since different amounts of resources were allocated for each task.

state. An experiment setting where the controls are highly constrained, and hence less straining to the simulator, is presented in Appendix G.2 and in Chung et al. [2020].

**Discussion of Physical Results**. These results are promising, not only from an algorithmic perspective, but also from a physics perspective. While there have been applications of machine learning techniques in nuclear fusion, they primarily focus on detecting disruptions and plasma instabilities [Cannas et al., 2013, Tang et al., 2016, Montes et al., 2019, Kates-Harbeck et al., 2019]. The work done by Baltz et al. [2017] is the closest to our application; however, since they were doing costly experiments online, their optimization leveraged human operators to ultimately decide which evaluation to perform next. To the best of our knowledge, our application is one of the first attempts in conducting offline optimization for tokamak control.

With these initial results established, we hope to continue progress on this problem by forming a closed loop controller for a tokamak. This requires expanding the number of tasks so that they cover the plasma's state space, and adapting to the fact that the state space is continuous (i.e. either via interpolation or using the continuous variant of MTS). Furthermore, we wish to develop more sophisticated plasma state representations, actions that can be applied, and reward functions in order to discover more interesting results. Lastly, readers may note that a controller derived from this method may not be optimal. Here, we have been seeking actions that myopically maximize reward; however, the real goal is to find an optimal *sequence* of actions that maximizes long term reward. We started with this approach since simulations are expensive, and we hope that this approximation still leads to a good controller. That being said, in the future we would like to extend the ideas of our algorithm to the reinforcement learning setting in order to derive sample efficient methods.

# 6    Conclusion

In this paper, we have proposed methods for dealing with many optimization problems that need to be solved simultaneously. We introduced a posterior sampling approach that has theoretic guarantees and often has dominant performance when compared to methods which do not distributed resources intelligently. This Thompson sampling method pairs nicely with our proposed locally stationary model, and we demonstrated that more sophisticated models are key when functions vary in difficulty. Finally, we used our algorithm to derive real results for nuclear fusion, which we hope to build upon in following work.

# 7    Acknowledgements

This material is based upon work supported by the National Science Foundation Graduate Research Fellowship Program under Grant No. DGE1252522 and DGE1745016. Willie Neiswanger is also

supported by NSF grants CCF1629559 and IIS1563887. Any opinions, findings, and conclusions or recommendations expressed in this material are those of the authors and do not necessarily reflect the views of the National Science Foundation.

Youngseog Chung is supported by the Kwanjeong Educational Foundation.

The authors would also like to thank the reviewers for their helpful feedback.

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
