[Supplementary Material · OfflineContextualBayesianOptimization_supplemental.pdf]

# Appendix

## A  Proof of Theorem 1

In this subsection we prove Theorem 1, and therefore assume that $\mathcal{X}$ and $\mathcal{A}$ are finite and that $t_{init} = 0$. We start by defining notation for the following analysis. Let $\mathcal{F}$ be the collection of all possible reward functions (i.e. the support of our prior). Recall that $D_T = \{(x_t, a_t, y_t)\}_{t=1}^T$ is the data sequence collected up to time $T$. Let $\mathcal{D}_T$ be the collection of all such $T$-length sequences, and note that the observations seen in this sequence depend on the reward function in question. We also denote $\mathcal{D} = \cup_{t=1}^\infty \mathcal{D}_t$. For $f \in \mathcal{F}$ and $D_T \in \mathcal{D}_T$, we define $\lambda(f, D_T)$ as

$$\lambda(f, D_T) := 1 - R_{T,f} \tag{5}$$

where $R_{T,f}$ is as defined in Theorem 1. Note that $\lambda(f, D_T) \geq 0$ for all $f \in \mathcal{F}, D_T \in \mathcal{D}_T$ and that $\max_{D \in \mathcal{D}} \lambda(f, D) = 1$ for all $f \in \mathcal{F}$. We define $D_{T,f}^*$ to be the optimal $T$-length data sequence with respect to $f$; that is,

$$D_{T,f}^* \in \underset{D_T \in \mathcal{D}_T}{\operatorname{argmax}} \lambda(f, D_T) \tag{6}$$

We will often write this sequence as $D_T^*$ when it is clear from context. Note that the optimal data sequence is also the one which makes greedy decisions at every time step. To better understand $D_T^*$, note that if $T = 1$, the evaluation selected will be the task and its corresponding best action that yields the greatest reward out of any task. If this optimal strategy is continued for more evaluations, each optimal task-action pair will be evaluated in order of descending reward, and after making $|\mathcal{X}|$ such evaluations, $\lambda(f, D_T^*) = 1$. After this, it does not matter which task-action pairs are evaluated.

Because the strategy of Algorithm 1 is to play myopically optimal with respect to a posterior sample, it falls into the broad class of algorithms known as *Myopic Posterior Sampling* (MPS) [Kandasamy et al., 2019a]. We first restate a known theorem and its conditions.

**Condition 1** ([Kandasamy et al., 2019a]). *Let $H \in \mathcal{D}$ be any arbitrary set of starting task-action evaluations. For reward functions $f, f' \in \mathcal{F}$ and corresponding optimal data sequences $D_{T,f}^*, D_{T,f'}^*$, there exists sequences $\{\epsilon_T\}_{T \geq 1}$ and $\{\tau_T\}_{T \geq 1}$ such that*

*1. The optimal data sequences achieve asymptotically similar performance:*

$$\sup_{f, f' \in \mathcal{F}} \sup_{H \in \mathcal{D}} \left\{ \mathbb{E}\left[\lambda(f, H \cup D_{T,f}^*)\right] - \mathbb{E}\left[\lambda(f', H \cup D_{T,f'}^*)\right] \right\} \leq \epsilon_T$$

*where the expectations are over the observed rewards.*

*2. The rate of convergence is better than $\mathcal{O}(1/\sqrt{T})$. That is, where $\sqrt{\tau_T} = 1 + \sum_{t=1}^T \epsilon_t$, we have that $\tau_T = o(T)$.*

**Condition 2** ([Kandasamy et al., 2019a, Golovin and Krause, 2011]). *Let $\mathbb{E}_{Y_{x,a}}$ denote the expectation over the likelihood $Y_{x,a} \sim \mathbb{P}(\cdot|x, a, f)$. Where $j < k$, let $D_j, D_k \in \mathcal{D}$ be such that $D_j$ is a prefix of $D_k$ (i.e. the first $j$ members of $D_k$ make up $D_j$). For all such $D_j, D_k \in \mathcal{D}, x \in \mathcal{X}, a \in \mathcal{A}$, and $f \in \mathcal{F}$ the following holds:*

*1. $\lambda$ is monotone, meaning that $\mathbb{E}_{Y_{x,a}}\left[\lambda(f, D_j \cup \{(x, a, Y_{x,a})\}\right] \geq \lambda(f, D_j)$.*

*2. $\lambda$ is adaptive submodular, meaning that,*

$$\mathbb{E}_{Y_{x,a}}\left[\lambda(f, D_j \cup \{(x, a, Y_{x,a})\})\right] - \lambda(f, D_j)$$
$$\geq \mathbb{E}_{Y_{x,a}}\left[\lambda(f, D_k \cup \{(x, a, Y_{x,a})\})\right] - \lambda(f, D_k)$$

Note that $\lambda$ as defined in (5) satisfies Condition 1. As mentioned before, the optimal strategy $\lambda(f, D_{T,f}^*) = 1$ for $T \geq |\mathcal{X}|$ and $\forall f \in \mathcal{F}$. This is true regardless of the initial data sequence $H$.

Therefore, we see that $\epsilon_T = 0$ for $T \geq |\mathcal{X}|$. Whenever $T < |\mathcal{X}|$, the largest $\mathbb{E}\left[\lambda(f, H \cup D_{T,f}^*)\right] -$

$\mathbb{E}\left[\lambda(f', H \cup D_{T,f'}^*)\right]$ can be is 1 since $\lambda$ is bounded. Therefore, we can set $\epsilon_T = 1$ for $T < |\mathcal{X}|$.

Putting this together,

$$\tau_T = \left(1 + \sum_{t=1}^{T} \epsilon_t\right)^2$$

$$\leq \left(1 + \sum_{t=1}^{|\mathcal{X}|-1} 1\right)^2$$

$$= (1 + |\mathcal{X}| - 1)^2 = |\mathcal{X}|^2$$

Thus Condition 1 holds with $\tau_T = |\mathcal{X}|^2$.

Our definition of $\lambda$ also meets the requirements of Condition 2. First of all, $\lambda$ is monotonically increasing because it relies on the maximum reward seen, and therefore $\lambda$ can only increase after seeing new data. $\lambda$ is also adaptive submodular since any improvement from seeing a new evaluation can only be more impactful when less data has been seen. For intuition, consider any evaluation $\{x, a, y\}$. There are two cases:

1. $\{x, a, y\}$ is better than any other evaluation for task $x$ in $D_k$. Therefore $D_j$ and $D_k$ will both have the same maximum played reward for task $x$; however, the previous maximum may have been greater for $D_k$ since it is a superset of $D_j$. Thus, the increase in $\lambda$ must be greater or equal for $D_j$.

2. $\{x, a, y\}$ is not best evaluation for task $x$ in $D_k$. In this case,

$$\lambda(f, D_k \cup \{(x, a, y)\}) - \lambda(f, D_k) = 0$$

**Theorem 2** ([Kandasamy et al., 2019a]). *Assume that $\lambda$ satisfies conditions 1 and 2, and let $\tau_T$ be as defined in Condition 1. Let $D_T$ be data collected by playing myopically optimal according to posterior samples. Then, for all $0 < \rho < 1$,*

$$\mathbb{E}\left[\lambda(f, D_T)\right] \geq (1 - \rho)\mathbb{E}\left[\lambda(f, D_{\rho T}^*)\right] - \sqrt{\frac{|\mathcal{X}||\mathcal{A}|\tau_T\gamma_T}{2T}}$$

Using this theorem, the proof for Theorem 1 is relatively straightforward.

***Proof** (Theorem 1).* Note that Theorem 2 can be used because Algorithm 1 optimizes $\lambda$ as defined in (5) with respect to posterior samples,

$$\mathbb{E}\left[\lambda(f, D_{\rho T}^*)\right] - \mathbb{E}\left[\lambda(f, D_T)\right] \leq \sqrt{\frac{|\mathcal{X}||\mathcal{A}|\tau_T\gamma_T}{2T}} + \rho\mathbb{E}\left[\lambda(f, D_{\rho T}^*)\right]$$

$$\leq |\mathcal{X}|\sqrt{\frac{|\mathcal{X}||\mathcal{A}|\gamma_T}{2T}} + \rho$$

In the case where $T > |\mathcal{X}|$, we can set $\rho = \frac{|\mathcal{X}|}{T}$. In this case,

$$\mathbb{E}\left[\lambda(f, D_{|\mathcal{X}|}^*)\right] - \mathbb{E}\left[\lambda(f, D_T)\right] = 1 - \mathbb{E}\left[\lambda(f, D_T)\right]$$

$$= \mathbb{E}\left[R_{T,f}\right]$$

$$\leq |\mathcal{X}|\sqrt{\frac{|\mathcal{A}||\mathcal{X}|\gamma_T}{2T}} + \frac{|\mathcal{X}|}{T}$$

If $T \leq |\mathcal{X}|$,

$$\mathbb{E}\left[R_{T,f}\right] \leq 1 < |\mathcal{X}|\left(\frac{1}{T} + \sqrt{\frac{|\mathcal{A}||\mathcal{X}|\gamma_T}{2T}}\right)$$

Therefore, the theorem holds. □

# B Synthetic Functions

## B.1 Branin-Hoo Function [Branin, 1972]

The Branin-Hoo maps $\mathbb{R}^2$ to $\mathbb{R}$, operates on the square $[-5, 10] \times [0, 15]$, and has the following form:

$$f(x) = a(x_2 - bx_1^2 + cx_1 - r)^2 + s(1 - t)\cos(x_1) + s$$

where $a = 1$, $b = \frac{5.1}{4\pi^2}$, $c = \frac{5}{\pi}$, $r = 6$, $s = 10$, and $t = \frac{1}{8\pi}$. It achieves a global minimum 0.397887 at three different values of $x$: $(-\pi, 12.275)$, $(\pi, 2.275)$, and $(9.42478, 2.475)$. Because we perform maximization in our experiments, we consider the negative Branin-Hoo function.

## B.2 Hartmann 4 and 6 Functions [Picheny et al., 2013]

The four dimensional version of the Hartmann function is evaluated on the domain $[0, 1]^4$. It has the form

$$f(x) = \frac{1}{0.839}\left[1.1 - \sum_{i=1}^{4} C_i \exp\left(-\sum_{j=1}^{4} a_{ji}(x_j - p_{ji})^2\right)\right]$$

The six dimensional Hartmann function is evaluated over $[0, 1]^6$ and has the form

$$f(x) = \frac{-1}{1.94}\left[2.58 + \sum_{i=1}^{4} C_i \exp\left(-\sum_{j=1}^{6} a_{ji}(x_j - p_{ji})^2\right)\right]$$

In both cases, $C$, $a$, and $p$ are defined as the following.

$$C = [1.0, 1.2, 3.0, 3.2]$$

$$a = \begin{bmatrix} 10.00 & 0.005 & 3.00 & 17.00 \\ 3.00 & 10.00 & 3.50 & 8.00 \\ 17.00 & 17.00 & 1.70 & 0.05 \\ 3.50 & 0.10 & 10.00 & 10.00 \\ 1.70 & 8.00 & 17.00 & 0.10 \\ 8.00 & 14.00 & 8.00 & 14.00 \end{bmatrix}$$

$$p = \begin{bmatrix} 0.1312 & 0.2329 & 0.2348 & 0.4047 \\ 0.1696 & 0.4135 & 0.1451 & 0.8828 \\ 0.5569 & 0.8307 & 0.3522 & 0.8732 \\ 0.0124 & 0.3736 & 0.2883 & 0.5743 \\ 0.8283 & 0.1004 & 0.3047 & 0.1091 \\ 0.5886 & 0.9991 & 0.6650 & 0.0381 \end{bmatrix}$$

These functions are typically used for minimization problems, so instead we use the negative versions.

## B.3 Independent Random Functions

In order to run experiments in which there are many different optimization problems, we consider drawing problems at random from a particular class of functions. These functions are over the domain $[0, 1]^d$ and have the following form:

$$f_{m,s,c,b}(x) = \sum_{k=1}^{m} \sum_{i=1}^{d} s_i \exp\left[-\frac{(x_i - c_{k,i})^2}{b_{k,i}}\right]$$

where $m$ is the number of point masses, $s_i$ are the scales, $c_{k,i}$ control the centers of the masses, and $b_{k,i}$ are the bandwidths. These quantities are generated at random in order to construct a suite of different functions. To ensure varying difficulty in the 30 randomly drawn functions, we drew 15 "easy" functions, 10 "average" functions, and 5 "hard" functions. These classes of difficulty are characterized by the possible values the parameters of $f$ can take (except for the centers of the mass, which can be anywhere in $[0, 1]^d$). To sample a function, each parameter is drawn uniformly at random over the support given in Table 1.

| Difficulty | Number of Masses | Scale Range | Bandwidth Range |
|---|---|---|---|
| Easy | $\{0, 1, 2\}$ | $[0, 1]$ | $[0.7, 0.9]$ |
| Average | $\{3, 4\}$ | $[0.25, 1]$ | $[0.4, 0.6]$ |
| Hard | $\{5, 6, 7\}$ | $[0.5, 1]$ | $[0.25, 0.4]$ |

Table 1: Ranges of parameters for randomly generated function by difficulty class.

## B.4 Correlated Tasks of Varying Difficulty

For correlated tasks of varying difficulty we use the function $g : \mathcal{X} \times \mathcal{A} \to \mathbb{R}$, where $\mathcal{X} = [0, 1]$ and $\mathcal{A} = [0, 1]$. From this, we discretize the problem by choosing ten equispaced tasks from $\mathcal{X}$. The function $g$ is defined as follows:

$$g(x, a) = \sum_{m=1}^{M} s_m \exp \left[ \frac{-(a - c_{a,m})^2}{b_{a,m}} + \frac{-((|x - 0.5| - c_{x,m})^2}{b_{x,m}} \right] \tag{7}$$

where the specific quantities are shown in the table below.

| $m$ | $s_m$ | $c_{a,m}$ | $b_{a,m}$ | $c_{x,m}$ | $b_{x,m}$ |
|---|---|---|---|---|---|
| 1 | 15 | 0.1 | 0.005 | 0.45 | 0.01 |
| 2 | $-15$ | 0.2 | 0.005 | 0.45 | 0.01 |
| 3 | 16 | 0.3 | 0.005 | 0.45 | 0.01 |
| 4 | $-6$ | 0.4 | 0.005 | 0.45 | 0.01 |
| 5 | 7 | 0.5 | 0.005 | 0.45 | 0.01 |
| 6 | $-7$ | 0.6 | 0.005 | 0.45 | 0.01 |
| 7 | 16 | 0.7 | 0.005 | 0.45 | 0.01 |
| 8 | $-16$ | 0.8 | 0.005 | 0.45 | 0.01 |
| 9 | 15 | 0.9 | 0.005 | 0.45 | 0.01 |
| 10 | 4 | 0.1 | 0.05 | 0.25 | 0.01 |
| 10 | $-4$ | 0.3 | 0.075 | 0.25 | 0.01 |
| 11 | 8 | 0.6 | 0.05 | 0.25 | 0.01 |
| 12 | $-4$ | 0.8 | 1 | 0.25 | 0.01 |
| 13 | 4 | 0.1 | 0.05 | 0.15 | 0.0025 |

Table 2: Parameters for (7)

## C  Risk Neutral Correlated Finite Task Experiments

In addition to the plots for correlated tasks shown in the second row of Figure 1, we also show risk-neutral versions of the same plots. That is, the same evaluations were made for each method, but instead of forming the policy estimate using best evaluations seen so far, we select the policy using best action according to the GP mean for each task. Ahead of time, we randomly select 10,000 actions uniformly at random for each of the tasks, and at every iteration we estimate which of these actions are best for their task based on the current GP posterior mean. We calculate regret with respect to the optimal policy for the pre-chosen set of actions.

We show risk-neutral policy performance for correlated tasks because REVI's goal is to increase the posterior mean as much as possible. That being said, Figure 4 shows that REVI does not significantly outperform MTS in most cases. Even in the Hartmann 4-2 example, where REVI performs best, the risk-averse MTS still performs best. Again, we hypothesize that REVI does not perform as well because the tasks for these experiments are sparse within the space. In the next appendix, we show that REVI in fact does well when every possible task in the space is considered (i.e. continuous tasks).

Figure 4: **Risk-neutral correlated tasks**. In order, the plots show the total simple regret for Branin 1-1 **(a)**, Hartmann 2-2 **(b)**, Hartmann 3-1 **(c)**, and Hartmann 4-2 **(d)**. The curves show the average over 10 trials, and the shaded region shows standard error.

# D  Continuous Multi-Task Thompson Sampling

We now consider the case in which the set of tasks is continuous. While it is impossible to make evaluations at every task in this case, we make the assumption that the tasks are correlated. As such, the joint task-action space can be modeled with a single GP, which can be used to estimate a mapping from task to action. Such a mapping can be found via a slightly modified version of MTS that we name Continuous Multi-task Thompson Sampling (CMTS). The intuition behind CMTS remains the same: at each iteration a finite subset of tasks is considered, a sample is drawn from the GP, and the task which sees the most improvement with respect to the sample is chosen. The only issue is that, for any task under consideration, there will be no previously collected data almost surely. As such, we suppose that the best action for the task is the action which achieves the greatest posterior mean. The final policy that we learn is simply the action that yields the greatest posterior mean for any given task (i.e. the risk-neutral strategy).

---

**Algorithm 2** Continuous Multi-Task Thompson Sampling (CMTS)

---

**Input:** capital $T$, initial capital $t_{init}$, mean function $\mu$, kernel function $\sigma$, and task subset size $s$.
Make random evaluations until $t_{init}$ evaluations are expended.
**for** $t = t_{init} + 1$ **to** $T$ **do**
    Draw $\widetilde{\mathcal{X}} \subset \mathcal{X}$ of size $s$.
    Draw $\widetilde{f} \sim GP(\mu, \sigma)|D_{t-1}$.
    Set $x_t = \underset{x \in \widetilde{\mathcal{X}}}{\operatorname{argmax}}\, \omega(x) \left[ \max_{a \in \mathcal{A}} \widetilde{f}(x, a) - \widetilde{f}\left(x, \underset{a \in \mathcal{A}}{\operatorname{argmax}}\, \mu_t(x, a)\right) \right]$
    Set $a_t = \underset{a \in \mathcal{A}}{\operatorname{argmax}}\, \widetilde{f}(x_t, a)$.
    Observe $y_t = f(x_t, a_t)$.
    Update $D_t = D_{t-1} \cup \{(x_t, a_t, y_t)\}$.
**end for**
**Output:** $\hat{h}(x) = \underset{a \in \mathcal{A}}{\operatorname{argmax}}\, \mu_T(x, a)$

---

Via experimentation, we found that, for a given task $x$, gauging improvement with respect to the greatest posterior mean value ($\max_{a \in \mathcal{A}} \mu_t(x, a)$) rather the value on the sample, $\widetilde{f}(x, \underset{a \in A}{\operatorname{argmax}}\, \mu_t(x, a))$, often works better in practice. Taking inspiration from Ginsbourger et al. [2014], we also cap the greatest posterior mean value by the greatest observation seen so far. We name this modified algorithm CMTS-PM (PM for "posterior mean"). Lastly instead of taking a sample over the entire space, we found that taking individual samples for each task is more computationally efficient without noticeably damaging performance.

For continuous tasks we compare against Profile Expected Improvement (PEI) [Ginsbourger et al., 2014] and the continuous version of REVI [Pearce and Branke, 2018]. Like CMTS, PEI picks whichever task is estimated to have the greatest improvement, but it estimates this using EI. In contrast to both CMTS and PEI, REVI selects points that are expected to yield the greatest improvement in

Figure 5: **Synthetic experiments for CMTS**. For each row from left to right, the plots show total simple regret for Branin 1-1, Hartmann 2-2, Hartmann 3-1, and Hartmann 4-2. The top row shows when the hyperparameters of the GP were estimated every iteration and the bottom shows when the hyperparameters were estimated beforehand.

posterior mean across all task. Although REVI's objective is better aligned with finding a policy that works well across all tasks, it can be much more expensive to compute. This is because the value of any candidate task-action pair must be estimated via a set of points in the task-action space. Further, it makes sense that this set should grow exponentially with dimension in order to get accurate estimates of a candidate's value.

We test these methods on the same functions as the correlated finite task case, but now with continuous tasks. Ahead of time, we select 250 tasks uniformly at random that policies will be tested against. For CMTS, CMTS-PM, and PEI 100 tasks and 100 actions are considered for every iteration. For specifics on REVI implementation details, see Appendix F. The GP for every experiment uses an RBF kernel, and hyperparameters were tuned by maximizing the marginal likelihood after every iteration. We also ran these same experiments but where hyperparameters were fixed ahead of time by sampling points uniformly at random and tuning according to marginal likelihood. The number of random samples for each experiment was equal to the total number of evaluations the algorithm could make. See Figure 5 for results.

When good estimates for hyperparameters are not known ahead of time (the most realistic case), we found that REVI performs well in low dimensions. Besides this, it seems that all of them methods have relatively similar performance in higher dimensions, with PEI and CMTS-PM possibly having a slight edge in some cases. It is possible that increasing the computational budget for REVI would result in better performance; however, REVI is already considerably slower than other methods (see Table 3). Interestingly, it seems that no method does significantly better than random for Hartmann 4-2. We believe that the main reason for this is that it is difficult to both pick point intelligently while simultaneously learning the hyperparameters for the model. Indeed, when hyperparameters are learned ahead of time, smarter methods again have an advantage over a random strategy for Hartmann 4-2. From these results, it also seems that REVI particularly seems to benefit from knowing hyperparameters ahead of time for most cases.

| Method | Time Elapsed (s) |
|---------|------------------|
| REVI | $175.14 \pm 0.57$ |
| Random | $36.80 \pm 0.25$ |
| PEI | $78.12 \pm 0.43$ |
| CMTS | $78.59 \pm 0.86$ |
| CMTS-PM | $78.74 \pm 0.49$ |

Table 3: **Run times for a single trial in Figure 5 (a)**. 10 trials were used to compute the average and standard deviation. Simulations were run on a desktop with Intel® Xeon(R) W-2123 CPU @ 3.60GHz and 16 GB of RAM.

# E   Algorithm Visualizations

In this subsection, we visualize the evaluations made by our MTS methods and REVI for the correlated task settings (Figure 6). Whereas our methods pick task-action pairs that end up clustering around where the true optimal policy is, REVI appears to pick pairs that outline the optimal policy and rarely picks evaluations in its immediate vicinity. This is the same behavior observed in Pearce and Branke [2018]. As a result, REVI does not perform as well in the boundary tasks for the finite case.

(a)

(b)

(c)

(d)

Figure 6: **Visualization of Methods.**. Each of the above plots show experiments on the Branin-Hoo function, where the x-axis shows varying task and the y-axis shows varying action. The plots on the left show when tasks are finite, and the plots on the right show the continuous case where all tasks in the space are considered. For each plot, 50 points were queried (shown in blue) to learn the policy. In the finite case, the optimal policy estimate is shown by stars, and in the continuous case the policy is drawn using a blue curve.

# F  More Implementation Details

## F.1  Acquisition Method Optimization

For each of the relevant acquisition methods (i.e. MTS, MEI, EI, TS, CMTS, CMTS-PM, PEI), we used random search to pick the next point. That is, for every iteration and every task, 100 actions are drawn uniformly at random to pick from (except for in Figure 2, in which 1,000 were used). Additionally, for the continuous methods, we pick 50 different tasks uniformly at random at which evaluations can be made every iteration.

## F.2  REVI Implementation Details

In every iteration, REVI considers a set of candidate points, and picks which one to make an evaluation at by gauging their impact via another set of judgement points. To form the candidate set in the finite task case, 100 actions are drawn uniformly at random for each task. A judgement set is formed in the exact same way for every iteration. In the continuous case, we choose 100 candidate points uniformly at random over the entire space for every iteration. Additionally, every iteration a grid of points are formed to judge the impact of every candidate point. We chose grid sizes of 50 tasks x 50 actions, 100 tasks x 100 actions, 100 tasks x 50 actions, and 100 tasks x 100 actions for experiments on Branin 1-1, Hartmann 2-2, Hartmann 3-1, and Hartmann 4-2, respectively. Tasks and actions are chosen uniformly at random, and the actions are the same for each task. Pearce and Branke [2018] suggest methods to more prudently select points for the judgement set; however, our attempts to implement these did not result in any benefit.

## F.3  Plotting Regret

The regret quantities that we use to plot are calculated in a number of different ways. If the true maximum is known (as is the case with Figure 1 (a)), then regret can be calculated easily. If there is a fixed grid of points over which the policy is evaluated over (i.e. in Figure 4 and Figure 5), then regret can be calculated with respect to the optimal policy for that grid. Otherwise, we take the greatest value found by any method, add a small amount to this value, and treat it as if it was the optimal. Although this is no longer a true notion of regret, it allows us to display performance of the methods in a more digestible manner.

# G  Nuclear Fusion Application

## G.1  Nuclear Fusion Experiment Details

We use the TRANSP program [Grierson et al., 2018] to simulate fusion reactions on DIII-D, a tokamak in San Diego that is operated by General Atomics. TRANSP is a time-dependent transport code used for interpretive analysis and predictive simulations of tokamaks. Access to TRANSP and running TRANSP experiments were possible thanks to our collaborators at Princeton Plasma Physics Lab. TRANSP operates by simulating real-world experiments (referred to as "shots") that were conducted on DIII-D. By running the predictive module of TRANSP, we are able to observe how changes in controls would affect the plasma. When simulating a given shot (a simulation on TRANSP is referred to as a "run"), we can identify variables at each time step that correspond to the state of the plasma. One such variable that we focus on is $\beta_n$, which is a ratio of the pressure of the plasma to the magnetic energy density. $\beta_n$ serves as a proxy for the economic output of the reaction. Besides this quantity, we also consider the *total energy eigenvalues*, which represent the amount of change in energy within and outside the plasma in response to certain perturbations. In particular, we focus on the minimum value of the total energy eigenvalues, which we will refer to as $\Delta\omega$. $\Delta\omega$ serves as a proxy for the stability of the plasma. Thus, in the context of our experiments, we assume that each state of the plasma can be fully characterized by a $(\beta_n, \Delta\omega)$ pair.

When conducting a simulation, we apply controls that specify parameters of the neutral beams, which include power, energy, full energy fraction and half energy fraction. The DIII-D tokamak has a total of 8 neutral beams, 6 of which are co-current beams (inject in the same direction as the plasma current) and 2 of which are counter-current beams (inject in the opposite direction of the plasma current). In our experiments, we confine the action space to 2 dimensions: power coefficient of

co-current beams and counter-current beams, each with domain [0.001, 1.0]. These power coefficients are applied by multiplying the maximum power of the set of beams by the coefficient. By ranging the power coefficient from 0.001 to 1.0, we essentially scale the beam powers from the minimum to the maximum power level possible.

We consider 7 distinct states of the plasma, which are represented by 7 shots from the DIII-D tokamak (the shots are 145699, 149689, 153145, 155215, 162939, 170473). In all 7 shots, a common instability called *tearing* occurred. Ideally, we would like to perform preventative measures once we sense a tear is about to occur. Therefore, we start the simulation 150 ms before time of tearing and run the simulator until 150 ms after the tearing. After the run completes, we extract the $\Delta\omega$ and $\beta_n$ values at 5ms increments throughout the duration of the run (total 300 ms) and take the median of them to produce $\overline{\Delta\omega}$ and $\overline{\beta}_n$. In order to balance between stability and the pressure in the tokamak, we set our reward to be $\overline{\beta}_n + 10\overline{\Delta\omega}$, where we chose coefficients based on the scales of each value. In summary, we optimize a combination of pressure and stability of the plasma, for each of the 7 different plasma states (7 contexts) simultaneously, by changing the power level of the co-current and counter-current beams (2D controls).

The optimization experiment results presented in Section 5 are averaged over 5 trials, each with 200 query capital. In each trial, for each task, 5 initial points are drawn uniformly at random for evaluation. Each task is modeled by a GP with an RBF kernel, and hyperparameters are tuned for a GP every time an observation is seen for its corresponding task by marginal likelihood.

Optimization was asynchronously parallelized with 10 workers. Kandasamy et al. [2018] proposed parallelized versions of standard Thompson sampling, and the algorithms used for the fusion experiment were the asynchronous Thompson sampling from Kandasamy et al. [2018] and the analougous parallel version of MTS.

### G.2    Alternative Experiment Settings and Results

(a)                                                          (b)

Figure 7: **Fusion Simulation Experiments under Different Experimental Settings**. In these experiments, the controls are constrained to be a multiplicative factor (less than 1.0) of the default controls for each shot. Each of the above show average values and standard error from 10 trials. **(a)** shows the total regret summed across all tasks and **(b)** regret achieved in each task. Note that curves differ in length for (b) since different amounts of resources were allocated for each task.

In Section 5 Figure 3, standard Thompson sampling (I-TS) outperforms MTS (I-MTS) when each state was modelled with an independent GP. Figure 3 (c) shows the allocation of queries across the states by MTS and TS. It is evident that MTS focuses on plasma state 3, even when there is little further improvement from the state. We believe that this is a systems issue where the simulator limits are tested with very high values for the neutral beam power levels. We conducted another set of optimization experiments with a different experimental settings which highly constrains the beam power levels and hence would be less straining to the simulator. In these experiments, instead of first

setting each beam to its maximum power level and then scaling it by a scalar between 0.001 and 1.0, we took the default power levels from the original shots and scaled these power levels by a scalar between 0.001 and 1.0. This ensures that beam powers will always be less than the original power levels. However, this limits querying the full range of power actually available from each beam. We tested I-MTS and I-TS under this change in settings and the results are presented in Figure 7. Note that we ran 10 trials of each algorithm with 125 timesteps per trial. We also have an additional plasma state from the shot 149205 from the DIII-D tokamak, which correspondes to plasma state 1. Lastly, the rewards components ($\overline{\beta}_n$ and $\overline{\Delta\omega}$) are scaled with different constants such that the reward is $10\overline{\beta}_n + 100\overline{\Delta\omega}$.

Under these settings, we can see that I-MTS outperforms I-TS. With stable outputs, I-MTS displays the expected behavior of selecting a plasma state that is deemed to provide most improvement and querying other states when reward has levelled off in a state (e.g. plasma state 4, 5).