[Reviews · NeurIPS 2019]

Reviewer 1



After reading the rebuttal I believe the authors can improve the quality of the paper with limited effort. I've updated my rating accordingly. ______________________________________________________ Originality: the paper follows the idea of previous works, but uses Thompson sampling in a novel way (to the best of my knowledge) for efficient task selection in contextual BO. However, the overall contribution of the paper I find less impactful, as it is a combination of known techniques (BO, Thompson sampling and regret minimization) and the context selection approach of previous work. The paper places itself well w.r.t. related work. Quality: the overall quality of the paper is good with an important exception. The kernel selection technique is well supported, the algorithm derivation is sound and we can also read about theoretical guarantees on minimizing overall task regret. However, as with many BO methods scaling to higher dimensional problems is an important aspect. Yet, the paper does not discusses this problem and the evaluations are rather low dimensional: 1-1 (context-action) dimensional for the synthetic data and 1-2 dimensional for the simulated nuclear fusion problem. Clarity: the paper overall reads well. However, I don't really understand why the paper stresses the 'offline' learning aspect. Not only the title emphasizes this, but the introduction puts lot of weight on this aspect. Yet, the contribution I feel has nothing to do with this problem. The efficient task selection aspect I think is an important requirement for any computational budget, not only for tasks with long simulation time (1 hour for the nuclear fusion). Based on the evaluation time the resulting BO technique can be applied offline/online, or anywhere in between. On another topic: while the paper describes the proposed algorithm well I don't really understand how the context parameters are represented for multiple tasks: is it a continuous, or a discrete variable? Can the algorithm choose from a fixed amount of contexts, or can it optimize it in a continuous setting? Given the tokamak example in the supplementary material it seems like we have a finite set of contexts to choose from, but the context variable itself is continuous. How do you interpolate? Significance: the paper at it present form needs some work in terms of evaluations and clarity to improve its significance. The main idea and the algorithmic approach may have some useful messages for practitioners, but the overall impact I find marginal.

Reviewer 2



This paper addresses the optimization of actions for a set of tasks, in a Bayesian optimization (BO) framework. The metamodels used are Gaussian processes, where to jointly model tasks and actions the use of a non-stationary parametric kernel is proposed. As for the infill criteria, Thomson sampling is proposed, with a theoretical analysis. In addition to several synthetic test cases, a real world application is considered with nuclear fusion in a tokamak. The paper is clear and well-written. It could be good to point out to recent overviews on BO, say, for instance: - Shahriari, B., Swersky, K., Wang, Z., Adams, R. P., & De Freitas, N. (2015). Taking the human out of the loop: A review of Bayesian optimization. Proceedings of the IEEE, 104(1), 148-175. - Frazier, Peter I. "A tutorial on Bayesian optimization." arXiv preprint arXiv:1807.02811 (2018). Thomson sampling is a local criterion, in the sense that it does not take into account the effect of one new design to reduce uncertainty on a given quantity of interest. Since the real objective is (1), Toscano-Palmerin, S., & Frazier, P. I. (2018). Bayesian optimization with expensive integrands. arXiv preprint arXiv:1803.08661 would be a pertinent comparator to add. Especially since it is not clear (P4L159) that the EI-based comparator matches with the references, or if it is a simplification. Section 4 There are many ways to encode non-stationarity between tasks (multi-output GPs, tasks modeled as categorical variables, warpings,…), is there a specific reason to pick the Gibbs kernel? Learning correlations between objectives is studied, e.g., in Shah, A., & Ghahramani, Z. (2016, June). Pareto frontier learning with expensive correlated objectives. In International Conference on Machine Learning (pp. 1919-1927), which shares the interest of learning from correlations. Minor remark: - P3L94: do you mean that those methods could not work with non-stationary kernels? #### Added after rebuttal #### I agree that the above reference of [Toscano-Palmerin et al.] is not exactly dedicated to the specific setup here, but once there is a GP over actions and tasks, adapting it is doable. Also, the non-stationary kernel is an orthogonal contribution with respect to the infill criterion, hence it cannot be used against other infill criteria. In my opinion, having less naive competitors will make the paper stronger and hence should be done; accordingly I kept the current score.

Reviewer 3



The authors introduce a novel Bayesian optimization approach based on Thompson sampling to to tackle multi-task optimization problem. Their approach is based on Gaussian Processes with Gibbs kernel which allows to capture the variation in difficulty between tasks; they use a Bayesian approach to learn the kernel hyperparameters. To the best of my knowledge, the authors cite and compare their work to the relevant papers for this task. However, I have not followed the recent developments of this field and can therefore not make a precise judgement regarding the advancements of the state of the art made by this submission. The paper seems technically sound. However, due to time constraints, I have not checked the proof of the theorem. The paper is clearly written and all the concepts are precisely described. The figures could be improved in order to be more easily readable when printed in B&W.

Reviewer 4



Originality: While the paper addresses an interesting problem, I don't understand how the definition of the problem in Section 3 differs from standard Bayesian optimization augmented with an additional input that describes the tasks. Even though in the experiments the authors compare to standard Bayesian optimization, where tasks are chosen uniformly at random, I don't see a reason why the task variable need to be treated differently in this setting, i.e why BO is not allowed to also select the task variable. I think modelling the task explicitly becomes more relevant if the cost of evaluating the blackbox function is taken into account as described by Swersky et al., such that one aims to select tasks and actions that not only gain the most information but are also cheap-to-evaluate. Quality: The paper provides a lower bound on the simple regret of their method and underpin their theoretical results by an empirical comparison to related work. However, in order to make the empirical results more convincing, the following points should be addressed in the experiment section: - a comparison of the proposed kernel to the MTBO task kernel proposed by Swerksy et al. which achieved state-of-the-art performance on these kind of benchmarks. - a comparison to standard Bayesian optimization where tasks are not sampled uniformly at random but instead are selected together with the input variable (i.e by optimizing the acquisition function) Clarity: In general the paper is well written and easy to follow. However, a few points should be addressed further: - using Thompson sampling as acquisition function requires to sample function values conditioned on the previous samples, or to put it differently, the function sample need to be sequentially constructed while being optimized. Are further approximations used (e.g see Lobato et al.) to avoid the cubic scaling of updating the Gaussian process? - What is the motivation of using a stationary task kernel described in Section 4? - Equation page 3 bottom: argmin -> argmax Significance: As explained above I don't fully understand why one cannot tackle this setting here with standard Bayesian optimization. Given that the difference in the fusion simulation experiments between the proposed method and standard Thompson sampling with randomly picked tasks seems to be relative small, I am afraid that the contributions of the paper are not sufficient for acceptance. Multi-task bayesian optimization Kevin Swersky, Jasper Snoek, Ryan P Adams Advances in neural information processing systems Predictive entropy search for efficient global optimisation of black-box functions José Miguel Hernández-Lobato, Matthew W Hoffman, Zoubin Ghahramani Advances in neural information processing systems ###### Post Rebuttal ######## I thank the authors for taking the time to respond to my review. The authors clarified the difference between standard BO where the input is augmented with task variables and their setting, however, a simple baseline that is missing, is to optimize one task at a time instead of sampling them randomly in each iteration. I will increase my score slightly (4 to 5), but still tend towards rejection.

[Author Response · NeurIPS 2019]

We would like to thank the reviewers for their valuable feedback. We have carefully thought through your comments.

- **Reviewer 1**

  Thank you for your review. We appreciate your point on demonstrating merit in higher dimensional settings. We have run additional synthetic experiments in higher dimensions since our submission that show our proposed algorithms outperform the comparison algorithms or are competitive with the best ones. We agree with your point that our algorithm can be applied in other situations besides the ones we emphasize. We meant for the term "offline" to differentiate our setting from the traditional one where context is decided by the state of nature. In our opinion, any situation in which nature can be bypassed and the context can be explicitly set could be considered offline. We will clarify this in the final paper.

  On whether the context variables are continuous or discrete, the context variables themselves can be either discrete or continuous variables, but we only consider the case where the number of contexts is finite. This is because the current algorithm requires observations to have been made in all contexts. One could make an alteration to the algorithm that leverages posterior mean rather than past observations in order to allow infinite contexts; however, we chose not to talk about that adaptation here.

- **Reviewer 2** Thank you for your review. We appreciate your suggestions for related work, and we will incorporate them into our final work. You suggested comparing to Toscano-Palmerin and Frazier's work in our experiments; however, we have reviewed these works and have noted that they are studying a different problem. Instead of finding the best action for each task, they find the best *single* action averaged over tasks.

  To address your questions about the Gibbs kernel, we picked this kernel because it was easy to choose the lengthscale function such that the lengthscales only changed as a function of the context/task variables. We did this to demonstrate that it is sometimes essential for lengthscales to change based on task (but not necessarily based on action). That being said, you are correct that there are probably other kernels that would allow us to avoid the problem of misjudging difficulty in the tasks.

  Regarding your last remark about previous methods' compatibility with non-stationary kernels, these methods are flexible to the choice of kernel; however, the authors chose to only use stationary kernels.

- **Reviewer 3**

  Thank you for your review. We will work on improving the figures to be more readable in black and white as well as our description of related methods.

- **Reviewer 4**

  Thank you for your review. You had questioned how our setting was different from standard Bayesian optimization and existing work in multi-task Bayesian optimization. What is crucial to note in our setting is that the optimal action may be different for each task. In particular, if one was to apply standard BO techniques on the joint task-action space, the algorithm would only identify a single optimal action. In contrast, our algorithm was designed to seek out the (most likely different) optimal actions corresponding to each task.

  This is also the key distinction between our work and that of Swersky et al. and Perrone et al. Their works aim to find a *single* action that either works well across all tasks or works well in a particular task. Therefore, these algorithms are not applicable here. Moreover, while having methods that trade off between cost and information gain makes sense in their setting, we believe that it does not make sense for our problem because one must adequately explore every task in order to find its corresponding best action.

  In response to your questions under the clarity section: it is true that approximations may be necessary to account for the cubic scaling of the GP; however, we did not require such approximations in our experiments, and such approximation methods are tangential to the goal of this paper. For the reasoning behind the Gibbs kernel, we wanted a locally stationary kernel (only stationary within a fixed task) in order to demonstrate the importance of having varying lengthscales as tasks change. You are correct in that there may be other non-stationary kernels that sufficiently model joint task-action space well, and we will expand on this point in the final paper.

[Meta-Review · NeurIPS 2019]

The contribution of this paper resides in the combination of known techniques (BO, Thompson sampling and regret minimization) to tackle a practical problem that had modest attention in the literature. The paper also provides theoretical guarantees on minimizing overall regret. I would encourage the authors to consider the outstanding concerns of the reviewers and to include the suggested round-robin baseline.